# Impact of AI-Based Post-Processing on Image Quality of Non-Contrast Computed Tomography of the Chest and Abdomen

**DOI:** 10.3390/diagnostics14060612

**Published:** 2024-03-13

**Authors:** Marcel A. Drews, Aydin Demircioğlu, Julia Neuhoff, Johannes Haubold, Sebastian Zensen, Marcel K. Opitz, Michael Forsting, Kai Nassenstein, Denise Bos

**Affiliations:** 1Institute of Diagnostic and Interventional Radiology and Neuroradiology, University Hospital Essen, Hufelandstraße 55, 45147 Essen, Germany; aydin.demircioglu@uk-essen.de (A.D.); johannes.haubold@uk-essen.de (J.H.); sebastian.zensen@uk-essen.de (S.Z.); marcel.opitz@uk-essen.de (M.K.O.); michael.forsting@uk-essen.de (M.F.); kai.nassenstein@uk-essen.de (K.N.); denise.bos@uk-essen.de (D.B.); 2Faculty of Medicine, University Duisburg-Essen, Hufelandstraße 55, 45122 Essen, Germany

**Keywords:** computed tomography, image reconstruction, image quality, chest, abdomen, urolithiasis, COVID-19, deep learning

## Abstract

Non-contrast computed tomography (CT) is commonly used for the evaluation of various pathologies including pulmonary infections or urolithiasis but, especially in low-dose protocols, image quality is reduced. To improve this, deep learning-based post-processing approaches are being developed. Therefore, we aimed to compare the objective and subjective image quality of different reconstruction techniques and a deep learning-based software on non-contrast chest and low-dose abdominal CTs. In this retrospective study, non-contrast chest CTs of patients suspected of COVID-19 pneumonia and low-dose abdominal CTs suspected of urolithiasis were analysed. All images were reconstructed using filtered back-projection (FBP) and were post-processed using an artificial intelligence (AI)-based commercial software (PixelShine (PS)). Additional iterative reconstruction (IR) was performed for abdominal CTs. Objective and subjective image quality were evaluated. AI-based post-processing led to an overall significant noise reduction independent of the protocol (chest or abdomen) while maintaining image information (max. difference in SNR 2.59 ± 2.9 and CNR 15.92 ± 8.9, *p* < 0.001). Post-processing of FBP-reconstructed abdominal images was even superior to IR alone (max. difference in SNR 0.76 ± 0.5, *p* ≤ 0.001). Subjective assessments verified these results, partly suggesting benefits, especially in soft-tissue imaging (*p* < 0.001). All in all, the deep learning-based denoising—which was non-inferior to IR—offers an opportunity for image quality improvement especially in institutions using older scanners without IR availability. Further studies are necessary to evaluate potential effects on dose reduction benefits.

## 1. Introduction

Obtaining excellent image quality on the one hand and reducing radiation exposure to the patients on the other hand is an ongoing challenge in modern computed tomography (CT) [1,2]. In this context, low-dose computed tomography (LD-CT) is an important diagnostic tool in clinical routine for detecting various pathologies including pulmonary infections or urinary stones [3,4]. Here, a lower radiation dose reduces the carcinogenic risk by ionizing radiation based on the assumption of the linear no-threshold model [5,6]. On the other hand, this leads to a weaker signal received by the image detector, which generally results in a lower image resolution [1]. However, the final image quality depends on the reconstruction algorithm applied to the raw image data.

A traditional method for image reconstruction is filtered back-projection (FBP), which is based on the idea that a projection consisting of measurements at multiple angles can be back-projected into a model of the scanned object using an inverse radon transformation with a high-pass filter [7]. The advantages of FBP are a high computational efficacy, stability, and speed in image reconstruction; while on the other hand, a high noise level and significant artifacts in low-contrast structures occur, especially in LD-CT. However, its advantages make it a widespread reconstruction method for old-generation CT scanners. Another modern approach is iterative reconstruction (IR), which requires more computational power and time but reduces image noise by cyclic imaging processing [8,9]. This leads to an overall better image quality, particularly in LD-CT, but can result in a plastic-like smooth appearance of the images [10].

With further increasing computational power within the last years, modern approaches to image noise reduction have been developed that are based on imaging post-processing including machine learning techniques [2,7,11]. A promising commercially available software solution is PixelShine v. 1.3.004 (AlgoMedica, Inc., Sunnyvale, CA, USA), which is based on a deep learning approach, trained by a high amount of imaging data [12].

First, clinical data in low-dose abdominal, pediatric thoracal, or midfacial trauma CT imply a significant noise reduction following PixelShine image reconstruction compared to FBP, but additional studies are necessary to evaluate its impact in different settings [13,14,15,16]. In our study, we evaluated the impact of the novel deep learning-based reconstruction algorithm on image quality in abdominal LD-CT for the detection of urinary stones as well as in chest CT for COVID-19 pneumonia diagnostics and compared it to FBP and IR.

## 2. Materials and Methods

### 2.1. Patients and Image Acquisition

Patient data were collected at two different sites of one radiological institution (site I (primary care): St. Marien-Hospital, Mülheim an der Ruhr, Germany; site II (tertiary care): University Hospital Essen, Essen, Germany). The institutional database of site I was searched for patients who underwent non-contrast thoracal between September 2020 and October 2021 for evaluation of pulmonary status for suspected COVID-19 pneumonia. Only examinations with the lowest available dose protocol were included. Because of the old scanner model (SOMATOM Emotion 6, Siemens Healthineers, Erlangen, Germany), this was not a real low-dose protocol as available in modern scanners. Twenty-nine patients were identified, of which three were excluded due to a different protocol (*n* = 2) and missing dose sheet (*n* = 1), leading to a final study population of 26 patients (8 female, 18 male) (Figure 1A). At site II, most thoracal CTs for suspected SARS-CoV-2 pneumonia were pulmonary artery CTs or full-dose thoracal CTs, mainly because of more severe disease or additional suspected pulmonary embolism. Therefore, no comparison was made between the sites.

Furthermore, the PACS of site I and institutional databases of site II were evaluated for patients with suspected urolithiasis who received low-dose, non-contrast abdominal CT between June 2019 and July 2021. A total of 48 patients from site I (17 female, 31 male) and 48 patients (17 female, 31 male) from site II were included (Figure 1B).

All scans (chest and abdomen) at site I were performed using SOMATOM Emotion 6 (Siemens Healthineers, Erlangen, Germany), while scans at site II were obtained using SOMATOM Force (Siemens Healthineers, Erlangen, Germany), a dual-source scanner which was used in single-source mode. Technical CT parameters were extracted from the DICOM header, and dose sheets were stored in the PACS and are summed up in Table 1. Raw image data of lung scans were reconstructed using FBP, using lung kernel (B70) and soft-tissue kernel (B40). Abdominal imaging was reconstructed using FBP (site I and site II) and IR with strengths 3 of 5 (site II only), applying a soft-tissue kernel. To reduce image noise, all reconstructed images (FBP and IR) were post-processed using the commercially available deep learning software PixelShine v. 1.3.004 (AlgoMedica Inc., Sunnyvale, CA, USA), resulting in PixelShine-processed FBP (FBP + PS) and IR (IR + PS) images. Following software reconstruction, parameters were used according to the manufacturer’s recommendations as follows: for the lung kernel, streak artifact reduction of R1, target noise level 12 (based on Hounsfield unit, HU), maximum strength of noise reduction A7, and lung kernel L6; for the soft-tissue kernel, streak artifact reduction of R2, sharpening P3, and target noise level 12 (based on Hounsfield unit, HU).

### 2.2. Objective Image Quality Analysis

For an objective analysis of thoracal and abdominal CTs, different regions-of-interest (ROIs) were evaluated (Figure 2). For thoracal CT, seven ROIs were defined as follows: ascending aorta (AAC), pulmonary trunk (TRP), descending aorta (ADE), lung (LUN), autochthonous back muscle (AUM), subcutaneous fat tissue (FAT), and air (AIR). Six ROIs were used for abdominal CT as follows: left liver lobule (LLL), spleen (SPL), descending aorta (AOD), autochthonous back muscle (AUM), subcutaneous fat tissue (FAT), and air (AIR). For each ROI, mean CT number and noise (SD of CT number as a surrogative parameter) were measured by evaluating a circular ROI (99.0 mm^2^ ± 1.18 [thorax] and 99.2 mm^2^ ± 1.22 [abdomen]). ROI annotation was performed on the FBP-reconstructed scan and was then automatically copied to each scan to ensure that all ROIs in each scan had the same annotation location. A custom tool written in Python was used for this.

Furthermore, the contrast-to-noise ratio (CNR) as well as the signal-to-noise ratio (SNR) were calculated using the following formulas [13]:(1)$$\;{CNR}_{ROI}=\frac{\left|{CT}_{ROI}-{CT}_{fat}\right|}{{SD}_{fat}}$$
(2)$$\;{SNR}_{ROI}=\frac{\left|{CT}_{ROI}\right|}{{SD}_{ROI}}$$

### 2.3. Subjective Image Quality Analysis

Three radiologists with 5 years (D.B., M.O.) and over 10 years (K.N.) of experience in CT independently evaluated the image quality of different methods for the reconstruction of thoracal as well as abdominal CT using a self-developed web-based application. In a blinded fashion, two differently reconstructed images showing the same slice of the same study were presented to the rater simultaneously in a randomized order (Figure 3 and Figure 4). Raters were able to vary the zoom level. Firstly, the rater had to rank the images using 1 for the better and 2 for the worse images. Equality could be expressed using 1 for both. Secondly, each image was rated with the help of a 5-point Likert scale (“excellent”, “completely acceptable”, “mostly acceptable”, “suboptimal”, and “unacceptable”). For thoracal CT, FBP and FBP + PS were compared for the lung as well as for the soft-tissue kernel. Comparisons of abdominal CTs were made between FBP and FBP + PS (site I + site II) as well as FBP + PS and IR (site II only) only for the soft-tissue kernel.

### 2.4. Statistical Analysis

The objective and subjective image quality of FBP + PS and FBP were compared for thoracal and abdominal CTs. An additional comparison of abdominal CTs between IR + PS and IR, as well as IR and FBP + PS, was performed for site II only, due to the unavailability of IR in site I. Mean and standard deviation (SD) or median and interquartile range (IQR)—in cases lacking a normal distribution—were used to report descriptive statistics. To compare demographics, Wilcoxon or *χ*2 tests were used. Concordance between subjective ratings were evaluated using Kendall’s coefficient of concordance W. A *p*-value < 0.05 was considered statistically significant. Statistical analysis was performed using R v4.2.0 and commercially available software (Microsoft Excel 2021, Redmond, WA, USA and SPSS 29.0, Inc., Chicago, IL, USA).

## 3. Results

### 3.1. Patient Cohort

Thoracal non-contrast CTs of 26 patients with suspected SARS-CoV-2 pneumonia (17 male, 9 females, median age 77.0 IQR: 16.5 years) and abdominal non-contrast LD-CTs of 48 patients (31 male, 17 female) at site I and site II, each suspected of urolithiasis, were evaluated (Table 2). For thoracal scans, median DLP was 301.4 mGy·cm (IQR: 188.6). For abdominal CT, median DLPs were 178.4 mGy·cm (IQR: 108.1) at site I and 60.0 mGy·cm (IQR: 40.2) at site II.

### 3.2. Objective Image Quality of Thoracal CT

Objective image quality was assessed by evaluating mean CT values and their standard deviation (as surrogate parameters for image noise) as well as SNR and CNR. Regarding thoracal CT, the mean difference of CT numbers between FBP + PS and FBP measured within the soft-tissue kernel varied between 1.11 HU (FAT) and −0.09 HU (AUM) and 1.98 HU (FAT) and 0.08 HU (ADE) for the lung kernel (Table 3). Significantly different CT values were detected only for lung, fat, and air independently of the kernel (<±2 HU). Noise (SD) was significantly reduced following PS processing in all ROIs in both kernels, with differences between −9.68 (ADE) and −2.33 (LUN) [lung kernel] and −28.98 (ADE) and −14.69 (FAT) [soft-tissue kernel]. Furthermore, all evaluated ROIs revealed significantly higher SNRs as well as CNRs in PS-post-processed images compared to FBP alone for both lung and soft-tissue kernels (Table 3 and Appendix A).

### 3.3. Objective Image Quality of Abdominal CT

The analysis of abdominal CTs delivered similar results when comparing FBP and FBP + PS in site I and site II (Table 4 and Appendix A). PS led to small differences in CT values which were again only significant for fat (1.00 ± 0.5 [site I] and 1.08 ± 0.6 [site II]) and air (0.83 ± 0.4 [site I] and 0.65 ± 0.5 [site II]). Noise was significantly reduced in all ROIs while SNR and CNR increased overall. Comparing IR + PS and IR, PS post-processing also decreased image noise and elevated SNR and CNR significantly, independently of the analysed ROI. Additionally, differences in CT values were low but significant not only for fat (0.92 ± 0.2) and air (0.98 ± 0.3) but also for the descending aorta (0.31 ± 0.4) and autochthonous muscle (0.10 ± 0.2). Finally, a comparison of FBP + PS and unprocessed IR images revealed that noise was also lower in FBP + PS while SNR was higher. Differences were significant for all ROIs except fat. No significant differences were seen with regard to CNR. Again, CT values were significantly altered for fat (1.15 ± 0.5) and air (1.68 ± 0.7).

### 3.4. Subjective Image Quality of Thoracal CT

Subjective image quality, comparing FBP + PS and FBP, was evaluated in thoracal CTs using the soft-tissue and lung kernels (Figure 3). All three raters preferred the PS post-processed images over the unprocessed ones for both the lung as well as the soft-tissue kernels. The agreement between the raters was moderate for the lung kernel (Kendalls’s W = 0.5, *p* = 0.01, Table 5) and perfect for the soft-tissue kernel (Kendalls’s W = 1, *p* < 0.001). The rating of the image quality on a 5-point Likert-scale could not deliver such a strong tendency. Regarding the lung kernel, two raters did not find any difference in image quality (mean difference = 0) while the third rated the FBP + PS just slightly better than FBP (mean difference 0.08). For the lung kernel, again one rater did not detect any quality difference while the others rated FBP + PS higher than FBP (mean differences 0.04 and 0.89, respectively). Interrater reliability was moderate for both kernels (Kendall’s W = 0.51 for soft tissue and 0.40 for lung) and was only significant for the soft-tissue kernel (*p* = 0.006).

### 3.5. Subjective Image Quality of Abdominal CT

In abdominal CTs, subjective image quality was compared using the soft-tissue kernel only (Figure 4A,B) and delivered similar results (Table 5). All raters strongly preferred FBP + PS over FBP (mean difference between 0.96 and 1) with an almost perfect interrater reliability for both sites (Kendall’s W = 0.98, *p* < 0.001).

In addition, a comparison betweem FBP + PS and IR was performed at site II (Figure 4C,D). Here, two raters ranked FBP + PS higher than IR (mean differences 0.44 and 0.79, respectively), while the third favoured IR over FBP + PS (mean difference −0.81), leading to a low interrater-reliability with a Kendall’s W of 0.19 (*p* = 1). Regarding the subjective quality rating, all observers strongly preferred FBP + PS over FBP (mean difference 0.63 to 0.96), with a significant interrater agreement (*p* < 0.001) (Table 6). Moreover, two observers preferred FBP + PS over IR (mean difference in rating: 0.5 and 0.19, respectively), while the third one did not show any preference. This resulted in moderate agreement, with a Kendall’s W of 0.44 (*p* = 0.02).

## 4. Discussion

In this retrospective analysis, the impact of AI-based post-processing of non-contrast chest and low-dose abdominal CT images was evaluated by comparing the commercial software PixelShine with unprocessed FBP and IR images in thoracal and abdominal CTs, showing the non-inferiority of PS post-processing compared to IR alone and therefore offering an opportunity for image quality improvement in institutions using older scanners without IR availability.

### 4.1. Objective Image Quality

Different CT reconstruction methods have various effects on image quality and noise. FBP is an efficient and stable approach but results in a high noise level and significant artifacts in low-contrast structures [17]. On the other hand, IR shows a better noise reduction but can result in a plastic-like smooth appearance of the images [7]. These differences in reconstruction techniques are foremost observed in chest and abdominal protocols [18]. The post-processing of FBP or IR images with PixelShine is advertised as reducing image noise but maintaining image information. However, published data on the impact of PixelShine on image quality are scarce.

As expected, image information was maintained using PixelShine as determined by changes in the CT values as a surrogative parameter. Small but significant alterations were only seen regarding fat, air, and lung tissue. This could be explained by their overall lower density and higher contrast between containing structures, which is affected by smoothing as it is caused by PixelShine more easily. However, since the differences were small (<±2 HU) and did not alter the visual impression, these are not considered clinically relevant, as similarly stated by Steuwe et al. [15].

Our study revealed an overall significant reduction of image noise on the one hand and a significant increase of CNR and SNR while maintaining image information for thoracal as well as abdominal non-contrast CT scans by post-processing FBP reconstructed images using PixelShine.

These findings were mostly independent of the applied kernel and the examined tissues/organs. These results are in line with previous reports. Brendlin et al. also stated that PixelShine improves the image quality of FBP-reconstructed thoracal LD-CT in paediatric patients, while Wisselink et al. described noise reduction, especially in air and emphysema, evaluating a COPDGene phantom [14,19]. But to our knowledge, our study is the first evaluation of PixelShine in adult chest CT in vivo—especially in the context of COVID-19 pneumonia. Regarding abdominal LD-CT, a study comparable to ours but with a lower number of patients was performed by Steuwe et al. and concluded that PixelShine reduces image noise and shows improved SNR and CNR [13]. Also, for other CT protocols including pelvic arterial phase CT or midfacial trauma imaging, image quality improvement was described by comparing FBP + PS and FBP resp. IR [16,20,21].

Importantly, we were also able to show that the image quality of PixelShine-processed images was at least not inferior to that of iteratively reconstructed abdominal CTs, as we found that noise and SNR were improved, while no significant differences were observed regarding CNR. Similar observations had earlier been made by Brendlin and Steuwe [13,14,15]. Although PixelShine is recommended by the vendor for quality improvement of FBP-reconstructed images, our additional analysis of PS-post-processed IR images also showed an optimization of CNR and SNR. But further studies are necessary to evaluate its clinical relevance. Therefore, additional subjective analyses are recommended and an evaluation of the plastic-like smooth appearance of IR images following PS-post-processing would especially be of interest [7].

### 4.2. Subjective Image Quality

Our subjective imaging evaluation confirmed the results from the objective analysis to a large extent. Imaging ranking as well as ranking on a 5-point Likert scale revealed a strong preference of the raters towards FBP + PS in abdominal LD-CT and in the thoracal CT image regarding the soft-tissue kernel. However, with regard to the lung kernel, no superiority of FBP + PS was seen. As this is the clinically relevant kernel in patients suspected of pneumonia, FBP + PS does not deliver a subjective benefit for these indications, which contrasts with observations made in paediatric chest CTs by Brendlin et al. [14]. But this could be explained by the patient selection, because Brendlin et al. evaluated ultra-low-dose CTs in children while we examined non-contrast CTs in adults with higher doses. Furthermore, the raters confirmed the non-inferiority of FBP + PS compared to IR-processed abdominal CT. This is important because modern CT scanners usually offer IR algorithms but there are still institutions with older machines where FBP is the only available reconstruction method; for these institutions especially, PixelShine could also offer a suitable opportunity for CT image quality improvement.

Another possible benefit of PixelShine is a further reduction of radiation exposure. Brendlin et al. could show, in a retrospective analysis of whole-body LD-staging CTs in melanoma patients using simulation software, that a dose reduction of up to 30% could be achieved without image quality loss using DL-based post-processing with PixelShine [22]. Although prospective data are not available yet, these findings underline the opportunities for dose reduction offered by DL-based post-processing software, which is especially important for commonly used protocols including thoracal and LD-abdominal CTs.

### 4.3. Limitations

We evaluated the usage of PixelShine in very narrowly defined CT indications (examination of suspected COVID-19 and urolithiasis), leading especially to a quite small population size for the thoracal imaging. This makes a generalization of our results difficult, although they are mostly in line with previous studies as described above. Due to the commercial character of the software, we had no insight into the detailed underlying algorithm, so our information on that fully relies on the vendor. A comparison with other commercial or non-commercial DL approaches such as ClariCT.AI (ClariPi, Seoul, Republic of Korea) [23,24,25], AiCE (advanced intelligent Clear-IQ Engine, Canon, Tokyo, Japan) [26,27,28], or TrueFidelity^TM^ (GE Healthcare, Chicago, IL, USA) would be favourable [29,30]. Moreover, we used the software’s parameters as recommended by the vendor. A detailed analysis comparing different software parameters was not feasible. In terms of image analysis, we assessed subjective image quality based on the raters’ overall impressions of the images and an additional ranking. No detailed analysis regarding the textural properties or visibility of artifacts was performed. Moreover, low-contrast features were not explicitly considered in our study as they were expected to play an overall minor role in low(er) dose CTs. Generally, a subjective assessment with a broader range of radiologists is recommended for future studies to evaluate its usability in clinical routines more specifically. With regard to future studies, a longitudinal follow-up of patients could be beneficial for investigating the impact of AI-based post-processing on clinical outcomes including diagnostic accuracy, changes in patient management, and potential dose reduction benefits.

In conclusion, we show that using PixelShine for deep learning-based post-processing of non-contrast thoracal and non-contrast, low-dose abdominal CT scans that were reconstructed using filtered back-projection leads to a significant increase in objective image quality and in subjective image quality, depending on the kernel in thoracal imaging and with the strongest effects in soft-tissue imaging. However, this post-processed image quality was not inferior to iterative reconstruction techniques, making it a promising approach for medical institutions where IR reconstruction methods are not available. Further studies are necessary to evaluate whether AI-based post-processing could allow for a further reduction of radiation dose while maintaining image quality.

## Figures and Tables

**Figure 1 diagnostics-14-00612-f001:**
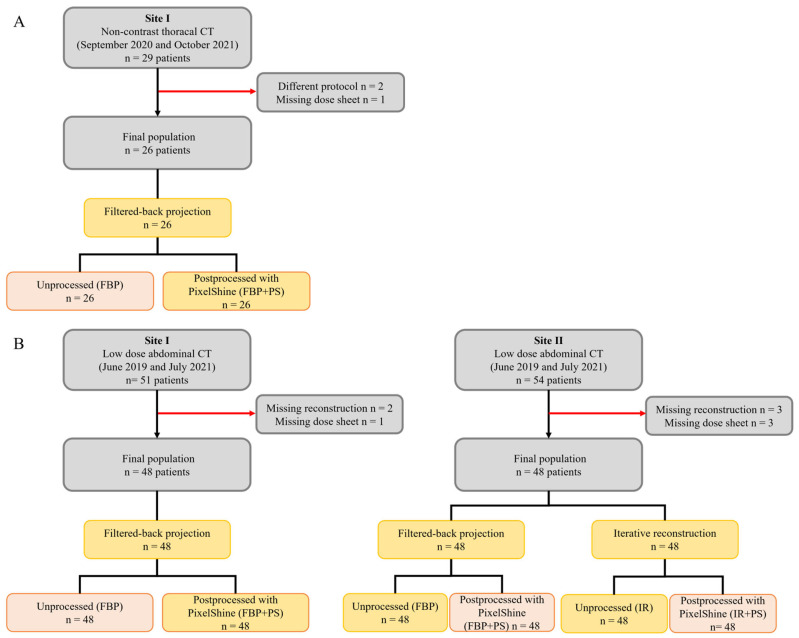
Patient flowchart and study design for assessment of thoracal (**A**) and abdominal LD-CT (**B**).

**Figure 2 diagnostics-14-00612-f002:**
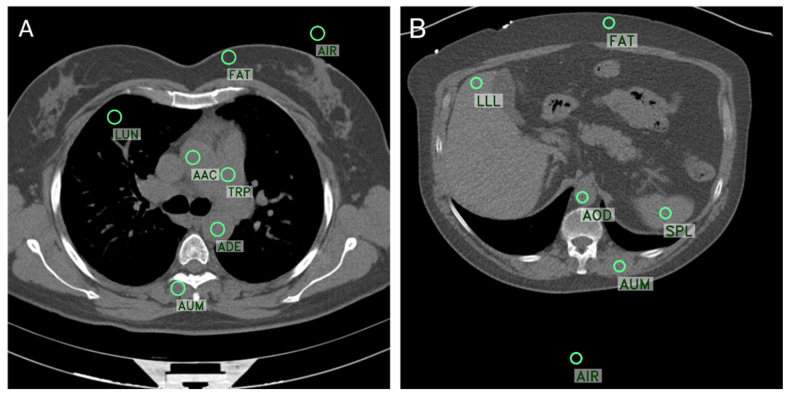
Axial CT slices reconstructed with FBP showing the positions of seven evaluated ROIs (green circle) for thoracal CT (**A**) and six ROIs for abdominal CT (**B**). AAC = ascending aorta, ADE = descending thoracal aorta, AOD = descending abdominal aorta, AIR = air, AUM = autochthonous muscle, FAT = subcutaneous fat, LLL = left liver lobule, LUN = lung tissue, TRP = pulmonary trunk, SPL = spleen.

**Figure 3 diagnostics-14-00612-f003:**
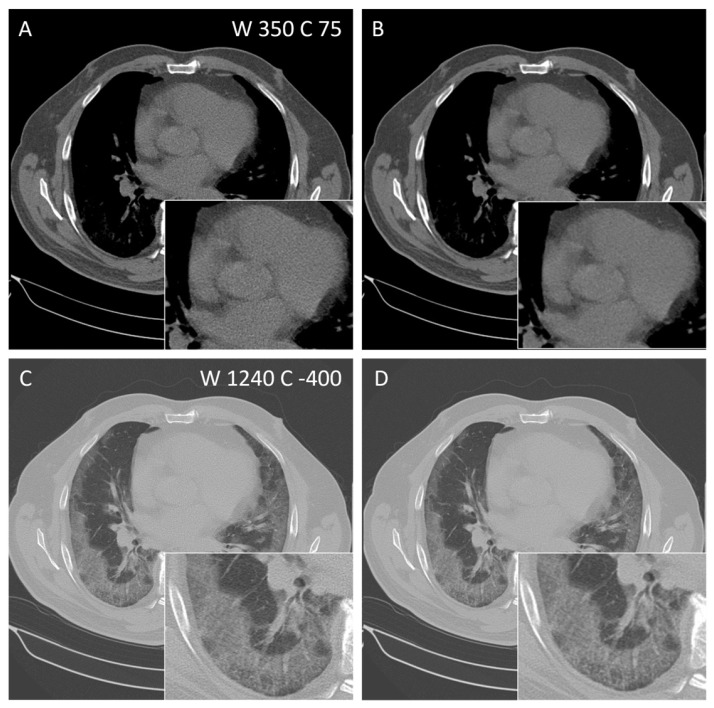
Axial CT of a 65-year-old patient with COVID-19 pneumonia comparing FBP reconstruction (**A**,**C**) and post-processing with PS (**B**,**D**) using soft-tissue (**A**,**B**) and lung kernel (**C**,**D**). Window width (W) and centre (C) are given.

**Figure 4 diagnostics-14-00612-f004:**
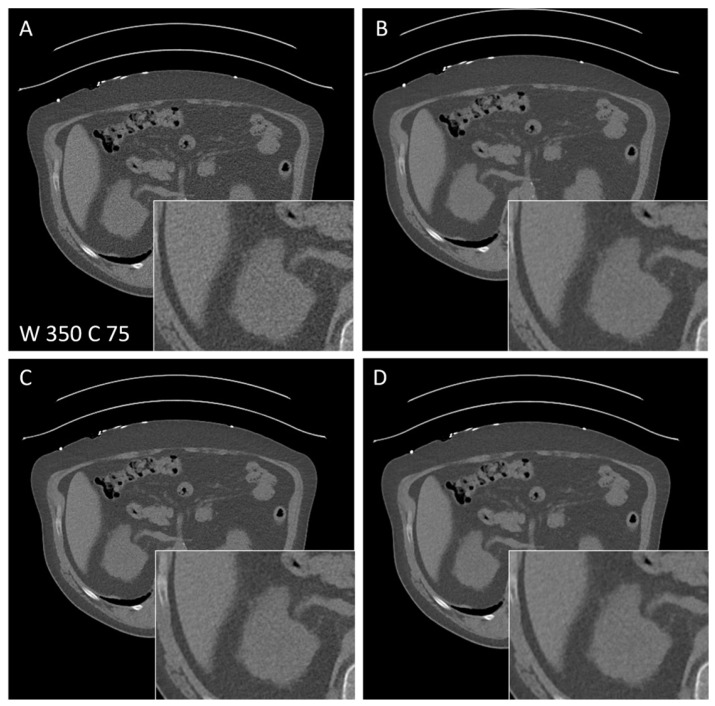
Axial CT of a 63-year-old patient suspected of urolithiasis comparing FBP reconstruction (**A**), FBP post-processed with PS (**B**), IR (**C**), and IR post-processed with PS (**D**). Window width (W) and centre (C) are given.

**Table 1 diagnostics-14-00612-t001:** Technical protocols for non-contrast thoracal and abdominal CT.

	Thorax	Abdomen	
	Site I	Site I	Site II
Scanner model	SOMATOM Emotion 6 (Siemens Healthineers)	SOMATOM Emotion 6 (Siemens Healthineers)	SOMATOM Force (Siemens Healthineers)
Scanner slices	6	6	192
Scanner mode	sequential	sequential	spiral
Tube potential (kVp)	130	130	150 or 100
Reference tube current time product (mAs)	70	40	40 or 244
Reconstructed slice thickness (mm)	2.5	5	5
Single collimation width (mm)	2	2	0.6
Total collimation width (mm)	12	12	57.6
Pitch factor	-	-	0.9

**Table 2 diagnostics-14-00612-t002:** Patient characteristics and radiation dose.

	Thorax	Abdomen	
	Site I	Site I	Site II
Patients	26 (9 female/17 male)	48 (17 female/31 male)	48 (17 female/31 male)
Median age (years)	77.0 (IQR 16.5)	55.5 (IQR 25)	39.0 (IQR 18)
Median tube Current time product (mAs) [Tube potential kVp]	86.5 (IQR 45) [130]	32.0 (IQR 18) [130]	220 (IQR 89.0) [100] 50 (IQR 25) [150]
Median scan length (cm)	32.1 (IQR 4.8)	47.6 (IQR 6.1)	49.6 (IQR 11.1)
Median CTDI_vol_ (mGy)	9.5 (IQR 4.9)	3.5 (IQR 1.9)	1.21 (IQR 0.6)
Median DLP (mGy·cm)	301.4 (IQR 188.6)	178.4 (IQR 108.1)	60.0 (IQR 40.2)

**Table 3 diagnostics-14-00612-t003:** Mean differences of CT values, noise, CNR, and SNR between FBP and FBP + PS reconstructed thoracal CT with lung and soft-tissue kernels. Significant *p*-values (<0.05) are in bold.

	Ascending Aorta	Pulmonary Trunk	Descending Aorta	Lung	Autochthonous Muscle	Fat	Air
B40 [soft-tissue kernel]							
CT FBP + PS vs. FBP	−0.08 ± 0.4 *p* = 0.116	−0.01 ± 0.4 *p* = 0.708	0.06 ± 0.5 *p* = 0.423	0.52 ± 0.5 ***p*** **≤ 0.001**	−0.09 ± 0.6 *p* = 0.94	1.11 ± 0.4 ***p*** **≤ 0.001**	0.79 ± 0.4 ***p*** **≤ 0.001**
Noise FBP + PS vs. FBP	−8.72 ± 3.4 ***p*** **≤ 0.001**	−9.28 ± 3.2 ***p*** **≤ 0.001**	−9.68 ± 3.4 ***p*** **≤ 0.001**	−2.33 ± 2.3 ***p*** **≤ 0.001**	−6.49 ± 2.1 ***p*** **≤ 0.001**	−4.10 ± 2.1 ***p*** **≤ 0.001**	−6.53 ± 2.6 ***p*** **≤ 0.001**
SNR FBP + PS vs. FBP	1.69 ± 0.6 ***p*** **≤ 0.001**	1.61 ± 0.5 ***p*** **≤ 0.001**	1.30 ± 0.5 ***p*** **≤ 0.001**	2.59 ± 2.9 ***p*** **≤ 0.001**	0.34 ± 0.3 ***p*** **≤ 0.001**	−2.11 ± 1.2 ***p*** **≤ 0.001**	-
CNR FBP + PS vs. FBP	2.99 ± 1.7 ***p*** **≤ 0.001**	2.96 ± 1.7 ***p*** **≤ 0.001**	2.92 ± 1.7 ***p*** **≤ 0.001**	15.92 ± 8.9 ***p*** **≤ 0.001**	2.60 ± 1.4 ***p*** **≤ 0.001**	-	-
B70 [lung kernel]							
CT FBP + PS vs. FBP	0.17 ± 0.6 *p* = 0.328	0.19 ± 0.5 *p* = 0.054	0.08 ± 0.7 *p* = 0.861	1.63 ± 0.8 ***p*** **≤ 0.001**	0.58 ± 0.9 ***p*** **= 0.002**	1.98 ± 0.4 ***p*** **≤ 0.001**	1.78 ± 0.6 ***p*** **≤ 0.001**
Noise FBP + PS vs. FBP	−27.95 ± 12.7 ***p*** **≤ 0.001**	−28.91 ± 12.0 ***p*** **≤ 0.001**	−28.98 ± 10.6 ***p*** **≤ 0.001**	−5.59 ± 8.3 ***p* = 0.001**	−20.60 ± 8.3 ***p*** **≤ 0.001**	−14.69 ± 8.9 ***p*** **≤ 0.001**	−14.93 ± 6.8 ***p*** **≤ 0.001**
SNR FBP + PS vs. FBP	0.46 ± 0.2 ***p*** **≤ 0.001**	0.44 ± 0.2 ***p*** **≤ 0.001**	0.38 ± 0.2 ***p*** **≤ 0.001**	1.63 ± 2.4 ***p* = 0.001**	0.14 ± 0.1 ***p*** **≤ 0.001**	−0.66 ± 0.46 ***p* ≤ 0.001**	-
CNR FBP + PS vs. FBP	0.95 ± 0.6 ***p*** **≤ 0.001**	0.94 ± 0.6 ***p*** **≤ 0.001**	0.93 ± 0.6 ***p*** **≤ 0.001**	5.22 ± 3.4 ***p*** **≤ 0.001**	0.83 ± 0.5 ***p*** **≤ 0.001**	-	-

**Table 4 diagnostics-14-00612-t004:** Mean differences of CT values, noise, SNR, and CNR comparing FBP + PS vs. FBP at site I and II and, additionally, IR + PS vs. IR and FBP + PS vs. IR reconstructed abdominal LD-CT at site II only. Significant *p*-values (<0.05) are in bold.

		Left Liver Lobule	Spleen	Descending Aorta	Autochthonous Muscle	Fat	Air
Site I							
FBP + PS vs. FBP	CT	0.05 ± 0.5 *p* = 0.532	−0.15 ± 0.7 *p* = 0.189	−0.05 ± 0.5 *p* = 0.579	−0.09 ± 0.5 *p* = 0.367	1.00 ± 0.5 ***p*** **≤ 0.001**	0.83 ± 0.4 ***p*** **≤ 0.001**
	Noise	−11.35 ± 5.3 ***p*** **≤ 0.001**	−8.67 ± 4.5 ***p*** **≤ 0.001**	−11.15 ± 4.6 ***p*** **≤ 0.001**	−7.00 ± 4.1 ***p*** **≤ 0.001**	−5.32 ± 3.5 ***p*** **≤ 0.001**	−6.80 ± 3.4 ***p*** **≤ 0.001**
	SNR	1.37 ± 0.5 ***p*** **≤ 0.001**	1.41 ± 0.5 ***p*** **≤ 0.001**	1.10 ± 0.4 ***p*** **≤ 0.001**	1.00 ± 0.5 ***p*** **≤ 0.001**	3.16 ± 2.32 ***p*** **≤ 0.001**	-
	CNR	4.68 ± 3.3 ***p*** **≤ 0.001**	4.51 ± 3.1 ***p*** **≤ 0.001**	4.46 ± 3.2 ***p*** **≤ 0.001**	4.31 ± 2.9 ***p*** **≤ 0.001**	-	-
Site II							
FBP + PS vs. FBP	CT	0.09 ± 0.7 *p* = 0.772	−0.11 ± 0.7 *p* = 0.291	0.26 ± 0.7 ***p*** **= 0.008**	−0.13 ± 0.6 *p* = 0.154	1.08 ± 0.6 ***p*** **≤ 0.001**	0.65 ± 0.5 ***p*** **≤ 0.001**
	Noise	−11.92 ± 2.8 ***p*** **≤ 0.001**	−9.98 ± 2.5 ***p*** **≤ 0.001**	−10.44 ± 2.9 ***p*** **≤ 0.001**	−8.25 ± 2.7 ***p*** **≤ 0.001**	−6.93 ± 2.3 ***p*** **≤ 0.001**	−6.55 ± 1.5 ***p*** **≤ 0.001**
	SNR	1.87 ± 0.5 ***p*** **≤ 0.001**	1.72 ± 0.6 ***p*** **≤ 0.001**	1.41 ± 0.5 ***p*** **≤ 0.001**	1.50 ± 0.6 ***p*** **≤ 0.001**	1.89 ± 0.9 ***p*** **≤ 0.001**	-
	CNR	3.12 ± 1.5 ***p*** **≤ 0.001**	3.01 ± 1.4 ***p*** **≤ 0.001**	2.80 ± 1.3 ***p*** **≤ 0.001**	3.03 ± 1.5 ***p*** **≤ 0.001**	-	-
IR + PS vs. IR	CT	0.10 ± 0.3 *p* = 0.06	0.01 ± 0.25 *p* = 0.495	0.31 ± 0.4 ***p*** **≤ 0.001**	0.10 ± 0.2 ***p*** **= 0.001**	0.92 ± 0.2 ***p*** **≤ 0.001**	0.98 ± 0.3 ***p*** **≤ 0.001**
	Noise	−4.03 ± 1.6 ***p*** **≤ 0.001**	−2.69 ± 1.3 ***p*** **≤ 0.001**	−3.32 ± 1.5 ***p*** **≤ 0.001**	−1.69 ± 1.3 ***p*** **≤ 0.001**	−1.13 ± 1.1 ***p*** **≤ 0.001**	−1.94 ± 1.3 ***p*** **≤ 0.001**
	SNR	1.00 ± 0.4 ***p*** **≤ 0.001**	0.72 ± 0.4 ***p*** **≤ 0.001**	0.73 ± 0.4 ***p*** **≤ 0.001**	0.47 ± 0.3 ***p*** **≤ 0.001**	−0.41 ± 0.5 ***p*** **≤ 0.001**	-
	CNR	0.71 ± 0.7 ***p*** **≤ 0.001**	0.69 ± 0.7 ***p*** **≤ 0.001**	0.65 ± 0.7 ***p*** **≤ 0.001**	0.70 ± 0.7 ***p*** **≤ 0.001**	-	-
FBP + PS vs. IR	CT	0.09 ± 0.6 *p* = 0.636	0.05 ± 0.6 *p* = 0.715	−0.09 ± 0.8 *p* = 0.367	−0.07 ± 0.5 *p* = 0.351	1.15 ± 0.5 ***p*** **≤ 0.001**	1.68 ± 0.7 ***p*** **≤ 0.001**
	Noise	−3.21 ± 1.9 ***p*** **≤ 0.001**	−1.96 ± 1.9 ***p*** **≤ 0.001**	−2.36 ± 1.9 ***p*** **≤ 0.001**	−0.90 ± 1.9 ***p*** **= 0.004**	0.20 ± 1.9 *p* = 0.373	−3.54 ± 1.3 ***p*** **≤ 0.001**
	SNR	0.76 ± 0.5 ***p*** **≤ 0.001**	0.55 ± 0.5 ***p*** **≤ 0.001**	0.48 ± 0.4 ***p*** **≤ 0.001**	0.27 ± 0.5 ***p*** **= 0.001**	0.05 ± 0.8 *p* = 0.221	-
	CNR	−0.01 ± 1.3 *p* = 0.378	−0.02 ± 1.3 *p* = 0.322	−0.03 ± 1.2 *p* = 0.278	−0.02 ± 1.3 *p* = 0.362	-	-

**Table 5 diagnostics-14-00612-t005:** Evaluation of subjective image quality ranking. Mean differences in ranking are shown for each rater (Raters 1–3) individually. Positive values indicate that the images of the first method were ranked higher. Interrater reliability was evaluated using Kendall’s W (W) and its *p*-value (*p*).

	Rater 1	Rater 2	Rater 3	W	*p*
Thorax					
B40 (FBP + PS vs. FBP)	1	1	1	1	<0.001
B70 (FBP + PS vs. FBP)	0.04	0.35	0.65	0.50	0.01
Abdomen (Site I)					
FBP + PS vs. FBP	0.96	1	1	0.98	<0.001
Abdomen (Site II)					
FBP + PS vs. FBP	1	0.96	1	0.98	<0.001
FBP + PS vs. IR	−0.81	0.44	0.79	0.19	1

**Table 6 diagnostics-14-00612-t006:** Evaluation of subjective image quality rating on a 5-point Likert scale. Mean ratings ± SD and mean differences in rating are shown for each rater (Raters 1–3) individually. Positive difference values indicate that the images of the first method were rated higher. Interrater reliability was evaluated using Kendall’s W (W) and its *p*-value (*p*).

	Rater 1	Rater 2	Rater 3	W	*p*
Thorax					
B40 (FBP + PS vs. FBP) Mean difference	4.0 ± 0 vs. 4.0 ± 0 0	4.0 ± 0 vs. 3.6 ± 0.5 0.38	4.0 ± 0.4 vs. 3.1 ± 0.3 0.88	1	<0.001
B70 (FBP + PS vs. FBP) Mean difference	4.0 ± 0 vs. 4.0 ± 0 0	4.1 ± 0.3 vs. 4.0 ± 0.2 0.08	3.4 ± 0.8 vs. 3.4 ± 0.8 0	0.50	0.01
Abdomen (Site I)					
FBP + PS vs. FBP Mean difference	4.4 ± 0.5 vs. 3.6 ± 0.5 0.80	4.3 ± 0.4 vs. 3.7 ± 0.6 0.63	4.2 ± 0.6 vs. 3.3 ± 0.5 0.92	0.59	<0.001
Abdomen (Site II)					
FBP + PS vs. FBP Mean difference	4.1 ± 0.4 vs. 3.2 ± 0.5 0.96	4.1 ± 0.3 vs. 3.3 ± 0.5 0.85	3.95 ± 0.3 vs. 3.0 ± 0.1 0.94	0.98	<0.001
FBP + PS vs. IR Mean difference	3.0 ± 0 vs. 3.0 ± 0 0	4.0 ± 0.2 vs. 3.8 ± 0.4 0.19	4.6 ± 0.6 vs. 4.1 ± 0.6 0.5	0.19	1

## Data Availability

Dataset available on request from the authors.

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
