# Peer review of "Impact of AI-Based Post-Processing on Image Quality of Non-Contrast Computed Tomography of the Chest and Abdomen"

_diagnostics, 2024, doi:10.3390/diagnostics14060612_

Round 1

Reviewer 1 Report

Comments and Suggestions for Authors

This paper conducted a clinical study comparing the image quality between FBP + AI-denoising vs IR and concluded that AI-based denoising, which is non-inferior to IR, offers an opportunity for image quality improvement. The paper did it through a series of objective and subjective studies.

The paper is well-written and well-organized. The studies are, in general, well-designed. However, I have a couple of questions that I hope to be clarified. How is the scanning dose selected? Did the authors consider the performance under low dose? How did the authors select the patients to be reviewed? Did the authors consider the patient's thickness and BMI? Did the authors consider low-contrast features?

Author Response

Dear reviewer, dear Ms. Chen

thank you very much for your response on our manuscript entitled “Impact of AI-based post-processing on image quality of non-contrast computed tomography of the chest and abdomen” and the opportunity for reconsideration after minor revision.

We revised the manuscript and considered your remarks as follows:

  1. How is the scanning dose selected?

The scanning dose is selected according to the specific protocol which is chosen with regard to the clinical indication (see question 2 and 3).

  1. Did the authors consider the performance under low dose?

Regarding the chest CT evaluation, only examinations with the lowest available dose protocol were included. Because of the old scanner type at Site I (SOMATOM Emotion 6, Siemens Healthineers) the overall dose of these examinations was higher compared to low-dose protocols of modern scanners. We clarified this in methods part 2.1. For evaluation of abdominal CTs, only low-dose examinations were included at both sites as stated in methods part 2.1, although again due to the older scanner at site I median CTDIvol and DLP were higher at site I (see table 2).

  1. How did the authors select the patients to be reviewed?

All patients who received a non-contrast chest CT for evaluation of pulmonary status for suspected COVID-19 pneumonia between September 2020 and October 2021 at Site I were included in the study. At site II, most CTs with this indication were full-dose and/or contrast-enhanced, mainly because of more severe disease or additional suspected pulmonary embolism, so no comparison was made between these sides. Regarding abdominal CT, all patients with suspected urolithiasis who received low-dose CTs at site 1 and 2 between June 2019 and July 2021 were included.  We clarified this in the methods part.

  1. Did the authors consider the patient's thickness and BMI?

Thank your very much for this remark. Patient’s thickness and BMI are important features that influence image quality and radiation dose. Unfortunately, due to the low population size (29 resp. 48 patients) no further patient selection or sub-group analysis regarding BMI and Patient’s thickness were performed.

  1. Did the authors consider low-contrast features?

(Objective) image quality assessment was based on the evaluation of the most important anatomic structures/tissues in the regarded regions. As chest as well as abdominal CTs were done using low(er) dose protocols an overall reduced discrimination of low-contrast features was expected and therefore not analysed in detail in our study. We have added a note on that in the limitations part of the discussion.

Yours sincerely

Marcel Drews

Reviewer 2 Report

Comments and Suggestions for Authors

Thank you for this interesting manuscript. The study evaluates the post-processing software PixelShine with its influence on CT value, noise, SNR, CNR and subjective image quality, based on CT images of the chest and abdomen. In total, data of 122 patients were analyzed. The application of PixelShine on filtered-backprojected images results in a lower image noise and improved image quality compared to filtered-backprojected images. In comparison to iterative reconstructions, differences were not remarkable.

General remarks:

·         The manuscript is well written but lacks several aspects

·         The topic has been evaluated beforehand but with a lower number of patients

·         No significant new results were presented

·         No clinical relevance was evaluated (e.g., visibility of certain structures)

·         The results are partly mixed up (site I vs. site II), with regard to the implementation of iterative reconstruction

Detailed remarks:

Abstract: Please include the main findings with numbers.

Introduction: short and concise.

Materials & Methods:

Why did you post-process IR-images? As far as I can read from the website, PixelShine is an alternative to iterative reconstruction, but not an additional post-processing method.

P4 Table 1: Were tube potentials fixed? Please provide the reference tube current for the corresponding tube potential, e.g., 40 mAs @ 150 kVp.

P4 L114: Were ROIs drawn within the same or multiple slices? Was the image quality of the python-based software identical to a typical PACS-system? Would there be a difference in image quality between the systems? Images came from two hospitals – were the image sets evaluated at the corresponding sites or stored and evaluated at one site?

P5 L131: Please elaborate on the analysis. Were the radiologists able to magnify the images? Did the radiologists evaluated the images based on the Likert-Scale in randomized order?  Please add window/levels for the different reconstructions (in the figures). Were radiologists able to change this during the subjective image analysis?

P6 L148: When did you provide mean and median values? Did you check for normal distribution? Usually, median values are provided if there is no normal distribution.

Results:

P6 L160: Why do you provide the mean value in the text but median value in the tables? Please unify.

P6 table 2: A median tube current time product does only make sense in the underlying tube potential is equal within the examinations. A mean tube current time product of 107.0 mAs with an IQR of 168.3 mAs suggests different underlying tube potentials. If you would like to provide the tube current time product, please differentiate between tube potentials.

P8 L181-192: Please use the same decimal places in the text and in the tables. Some numbers are undetectable in the tables (e.g., 1.00+/-0.29). Please check all numbers.

P9 L208-220: Please provide the ratings of the radiologists. The differences are interesting, but the underlying ratings are required as well.

P9 L227/table 5+6: You mention that only site II has iterative reconstruction. The text and table describe something else. Please check thoroughly.  

Discussion:

How does your study differ from previously studies, except for the larger study cohort? Please clarify this point more specifically.

P10 L262: What do you mean by the sentence “These differences in reconstruction techniques are foremost observed in protocols [18]”. What protocols are meant?

What impact does PixelShine have on IR-processed images? What happens with a plastic-like smooth appearance after applying PixelShine?

Author Response

Dear reviewer, dear Ms. Chen

thank you very much for your response on our manuscript entitled “Impact of AI-based post-processing on image quality of non-contrast computed tomography of the chest and abdomen” and the opportunity for reconsideration after minor revision.

We revised the manuscript and considered your remarks as follows:

  1. Abstract: Please include the main findings with numbers.

Numbers have been added to the abstract.

Materials & Methods:

  1. Why did you post-process IR-images? As far as I can read from the website, PixelShine is an alternative to iterative reconstruction, but not an additional post-processing method.

Thank you for this question. We agree, that PixelShine is intended to improve image quality of FBP-reconstructed images as an alternative for iterative reconstruction. But we aimed to test, if PixelShine would be able to improve IQ of IR-images even further, which is shown in table 4. However, no subjective IQ analysis was performed to evaluate clinical relevance of these results.

  1. P4 Table 1: Were tube potentials fixed? Please provide the reference tube current for the corresponding tube potential, e.g., 40 mAs @ 150 kVp.

Tube potentials were fixed as given in table 1.

  1. P4 L114: Were ROIs drawn within the same or multiple slices? Was the image quality of the python-based software identical to a typical PACS-system? Would there be a difference in image quality between the systems? Images came from two hospitals – were the image sets evaluated at the corresponding sites or stored and evaluated at one site?

As described in methods part 2.2 and shown in figure 2, the ROIs were drawn within the one FBP-reconstructed slice and automatically copied to the other reconstructions to ensure same annotation location. The DICOM data of all studies from both sites were imported into the Python-based software and evaluated. No differences in image quality compared to normal PACS software were observed.

  1. P5 L131: Please elaborate on the analysis. Were the radiologists able to magnify the images? Did the radiologists evaluated the images based on the Likert-Scale in randomized order?  Please add window/levels for the different reconstructions (in the figures). Were radiologists able to change this during the subjective image analysis?

The images were presented to the raters in a randomized order. Zoom level could be varied while the windows were fixed. We clarified this in the methods part and added window levels to the figures.

  1. P6 L148: When did you provide mean and median values? Did you check for normal distribution? Usually, median values are provided if there is no normal distribution.

Due to lacking normal distribution of patient data (table 2), Median values were chosen. A remark was added in the methods part.

Results:

  1. P6 L160: Why do you provide the mean value in the text but median value in the tables? Please unify.

Thank you, this mistake was corrected.

  1. P6 table 2: A median tube current time product does only make sense in the underlying tube potential is equal within the examinations. A mean tube current time product of 107.0 mAs with an IQR of 168.3 mAs suggests different underlying tube potentials. If you would like to provide the tube current time product, please differentiate between tube potentials.

Thank you for this remark, we differentiated between the tube potentials now.

  1. P8 L181-192: Please use the same decimal places in the text and in the tables. Some numbers are undetectable in the tables (e.g., 1.00+/-0.29). Please check all numbers.

Thank you, this has been corrected.

  1. P9 L208-220: Please provide the ratings of the radiologists. The differences are interesting, but the underlying ratings are required as well.

Thank you again for this interesting remark. We added the ratings in table 6.

  1. P9 L227/table 5+6: You mention that only site II has iterative reconstruction. The text and table describe something else. Please check thoroughly.  

This mistake has been corrected.

Discussion:

  1. How does your study differ from previously studies, except for the larger study cohort? Please clarify this point more specifically.

An additional remark highlighting our new results regarding the use of PixelShine in adult chest CTs in context of COVID-19 pneumonia was added.

  1. P10 L262: What do you mean by the sentence “These differences in reconstruction techniques are foremost observed in protocols [18]”. What protocols are meant?

Thank you, the missing words “chest and abdominal” have been added.

  1. What impact does PixelShine have on IR-processed images? What happens with a plastic-like smooth appearance after applying PixelShine?

Objective image quality analysis revealed improved CNR and SNR following PS-post-processed IR images as stated in the results part. Additional subjective analysis was not explicitely done, but is now recommended to evaluate the effect of subjective image quality, especially regarding plastic-like smooth appearance. We added this point to the discussion  due to your helpful remark.

We hope that the reworked manuscript does now meet the your satisfactions and will be reconsideration for publication.

Yours sincerely

Marcel Drews

Reviewer 3 Report

Comments and Suggestions for Authors

Summary:

The research article titled "Impact of AI-based post-processing on image quality of non-contrast computed tomography of the chest and abdomen" evaluates the efficacy of a deep learning-based software, PixelShine, in improving the image quality of non-contrast chest and abdominal CT scans. The study is retrospective, comparing the objective and subjective image quality of images reconstructed using traditional methods (filtered back-projection (FBP) and iterative reconstruction (IR)) with those post-processed by PixelShine. The findings demonstrate significant noise reduction and maintenance of image information with AI-based post-processing, suggesting potential benefits in soft-tissue imaging and for institutions using older scanners without IR capability.

General Strengths:

  1. Comprehensive Evaluation: The study thoroughly compares traditional image reconstruction techniques with advanced AI-based post-processing, providing valuable insights into the potential of deep learning in radiology.
  2. Objective and Subjective Analyses: Incorporating objective metrics and subjective assessments by experienced radiologists adds depth to evaluating image quality improvements.

Weaknesses:

  1. Limited Generalizability: The study is conducted retrospectively with a specific focus on non-contrast CT scans for suspected COVID-19 pneumonia and urolithiasis, which may limit the applicability of the findings to other types of CT scans or conditions.
  2. Lack of Comparison with Other AI Methods: Only one commercial AI software is tested without comparison to other AI-based post-processing tools, potentially limiting the understanding of PixelShine’s relative performance.
There are some minor points I would suggest the authors to revise. I anticipate some of these points will be hard to address in the setting of this study, but they should be discussed in greater detail nonetheless:
  1. Broader Evaluation: Future studies should include a wider range of CT applications and compare the performance of PixelShine against other AI-based post-processing tools to better understand its relative strengths and weaknesses.
  2. Longitudinal Follow-up: Investigate the impact of AI-based post-processing on clinical outcomes, including diagnostic accuracy, patient management changes, and potential dose reduction benefits.
  3. Technical Details: Provide more information on the AI model’s training data, including diversity and volume, to assess the robustness and generalizability of the algorithm.
  4. User Experience Feedback: Include feedback from a broader range of radiologists to understand the clinical usability and integration of AI-based post-processing in routine practice.

Author Response

Dear reviewer, dear Ms. Chen

thank you very much for your response on our manuscript entitled “Impact of AI-based post-processing on image quality of non-contrast computed tomography of the chest and abdomen” and the opportunity for reconsideration after minor revision.

We revised the manuscript and considered your remarks as follows:

  1. Broader Evaluation: Future studies should include a wider range of CT applications and compare the performance of PixelShine against other AI-based post-processing tools to better understand its relative strengths and weaknesses.

Thank you for highlighting this important point which we considered in the discussion (Ll.  328 ff.)

  1. Longitudinal Follow-up: Investigate the impact of AI-based post-processing on clinical outcomes, including diagnostic accuracy, patient management changes, and potential dose reduction benefits.

We included this helpful remark as a recommendation for future studies in the discussion.

  1. Technical Details: Provide more information on the AI model’s training data, including diversity and volume, to assess the robustness and generalizability of the algorithm.

Because PixelShine is a commercial product, unfortunately detailed information on the models training data is not available. We mentioned this in the discussion part.

  1. User Experience Feedback: Include feedback from a broader range of radiologists to understand the clinical usability and integration of AI-based post-processing in routine practice.

Thank you again for this interesting remark which we have no added  to the discussion.

We hope that the reworked manuscript does now meet the your satisfactions and will be reconsideration for publication.

Yours sincerely

Marcel Drews